# VLCKD in Combination with Physical Exercise Preserves Skeletal Muscle Mass in Sarcopenic Obesity after Severe COVID-19 Disease: A Case Report

**DOI:** 10.3390/healthcare10030573

**Published:** 2022-03-19

**Authors:** Elisabetta Camajani, Alessandra Feraco, Sabrina Basciani, Lucio Gnessi, Luigi Barrea, Andrea Armani, Massimiliano Caprio

**Affiliations:** 1PhD Program in Endocrinological Sciences, University of Rome “La Sapienza”, 00161 Rome, Italy; elisabetta.camajani@uniroma1.it; 2Department of Human Sciences and Promotion of the Quality of Life, San Raffaele Roma Open University, 00166 Rome, Italy; alessandra.feraco@sanraffaele.it (A.F.); andrea.armani@sanraffaele.it (A.A.); 3Laboratory of Cardiovascular Endocrinology, IRCCS San Raffaele Roma, 00166 Rome, Italy; 4Department of Experimental Medicine, University of Rome “La Sapienza”, 00161 Rome, Italy; sabrinabasciani@yahoo.it (S.B.); lucio.gnessi@uniroma1.it (L.G.); 5Endocrinology Unit, Department of Clinical Medicine and Surgery, University Federico II, 80131 Naples, Italy; luigi.barrea@unina.it

**Keywords:** physical training, ketogenic diet, SARS-CoV-2, ketone bodies, obesity, sarcopenia

## Abstract

The prevalence of sarcopenic obesity is increasing worldwide, with a strong impact on public health and the national health care system. Sarcopenic obesity consists of fat depot expansion and associated systemic low-grade inflammation, exacerbating the decline in skeletal muscle mass and strength. Dietary approach and physical exercise represent essential tools for reducing body weight and preserving muscle mass and function in subjects with sarcopenic obesity. This case report describes the effects of a dietary intervention, based on a Very-Low-Calorie Ketogenic Diet (VLCKD) combined with physical exercise, on body composition, cardiometabolic risk factors, and muscle strength in a woman with sarcopenic obesity, two weeks after hospitalization for bilateral interstitial pneumonia due to COVID-19. To our knowledge, this is the first case report to describe the efficacy of a combined approach intervention including VLCKD along with physical exercise, in reducing fat mass, improving metabolic profile, and preserving skeletal muscle performance in a patient with obesity, soon after severe COVID-19 disease.

## 1. Introduction

Sarcopenic obesity (SO) is a clinical condition characterized by visceral fat accumulation and is associated with reduced skeletal muscle mass [1,2,3]. Sarcopenia is a term used to describe a progressive decline in skeletal muscle mass and strength occurring with aging, and is associated with significant morbidity and mortality [4,5,6]. Recently, the European Working Group on Sarcopenia in Older People (EWGSOP2) published a consensus paper providing specific criteria for sarcopenia identification and characterization in clinical practice, and it indicated that poor muscle function is a major determinant for sarcopenia development [4]. Muscle mass, strength, and physical performance represent the measurable readouts that define sarcopenia. According to the EWGSOP2, low muscle strength, measured by the handgrip strength test or chair stand test, is the primary parameter of sarcopenia; muscle strength is presently the most reliable measure of muscle function. Sarcopenia diagnosis is confirmed by the presence of low muscle quantity (evaluated by dual-energy X-ray absorptiometry or bioelectrical impedance analysis) or quality. In aging subjects, visceral fat accumulation is often combined with muscle loss, due to obesity-induced low-grade inflammation, leading to muscle wasting and sarcopenia development [7,8]. In particular, muscle and strength loss in women after menopause is the result of multiple factors, which are mostly dependent on physical inactivity, malnutrition, mitochondrial stress, systemic inflammation, and hormonal changes that can also contribute to obesity. Accordingly, overweight/obese patients with metabolic syndrome display muscle insulin resistance, which is also due to lipotoxicity induced by ectopic lipid deposition. In turn, this promotes a loss of strength and muscle mass, as well as a reduced functional capacity, which is associated with aging [9,10,11,12]. On the other hand, sarcopenia incidence in obesity is underestimated due to the scarce availability of accurate body composition assessment techniques, thus suggesting that in obese individuals, a reduction in muscle strength and function can occur without any evidence of a reduction in muscle mass [13]. Lifestyle modifications with adequate nutrition and proper physical activity are essential to counteract SO progression [14,15]. An elevated body mass index (BMI), as well as obesity-related metabolic alterations, represent important risk factors for complications and mortality following SARS-CoV-2 infection [16]. In particular, several clinical studies indicated hyperglycemia as a predictor of COVID-19 fatalities [17,18,19,20]. The ketogenic diet was considered as an effective nutritional strategy during the COVID-19 pandemic [21], and the use of digital platforms has proven to be extremely effective in the management of patients requiring medical and/or nutritional support, as they encourage adherence to dietary- and/or exercise-based interventions, which can potentially lead to long-term weight loss and maintenance [22]. For the first time, this case report provides evidence for the efficacy of a combined approach intervention including a VLCKD along with interval training in reducing fat mass, improving metabolic profile, and preserving the skeletal muscle performance of a female subject after hospitalization for severe COVID-19. 

## 2. Case Description

A 55-year-old post-menopausal woman, affected by second-degree obesity, hypertension, hyperinsulinemia, hypercholesterolemia, hypertriglyceridemia, and sarcopenia, was admitted in May 2021 at the Center of High Specialization for the Care of Obesity, Sapienza University of Rome, Italy. Upon admission, the patient signed an informed consent form in accordance with the General Data Protection Regulation (GPDR, 2016/679). In March 2021, the patient tested positive for COVID-19 and was hospitalized in a subintensive care unit with bilateral interstitial pneumonia from 2 April2021 until 16 April2021. During hospitalization, heparin, dexamethasone, and a high-flow nasal cannula (HFNC) were administered. On 5 May 2021, the patient tested negative for COVID-19. After discharge, home therapy involved taking pantoprazole 40 mg, olmesartan medoxomil 20 mg, and methylprednisolone 16 mg for three days only, and dexamethasone 4 mg/mL for five days only. This patient was screened from 24 May to 16 July2021, and was prescribed a VLCKD with meal replacements (800 kcal/day) for six weeks, with the following composition of macronutrients as a relative percentage of caloric intake: carbohydrates 28 g (14.6%), olive oil 20 g plus 15 g of lipids from other sources (38.7%), and proteins 85 g (46.7%). She consumed four meal replacements per day (the timings of the main meals were at 8 a.m., 1.00 p.m., 8.00 pm, and mid-afternoon), which contained whey and vegetable proteins derived from soya, green peas, or cereals, and one serving of vegetables with a low glycemic index at lunch and dinner. Supplements of vitamins, minerals and omega-3 fatty acids were provided in accordance with international recommendations (EFSA 2017). It was also recommended to drink at least 2–2.5 L of water per day. In accordance with the Position Statement of the Italian Society of Endocrinology (SIE) [23], this patient was closely and periodically monitored through physical examination (anthropometric measurements, blood pressure (BP), heart rate, body composition parameters) and biochemical analyses. Blood tests (blood count, electrolytes, glucose, insulin, lipids, total proteins and albumin, plasma creatinine, blood urea nitrogen and uric acid, alanine transferase, aspartate transaminase, and estimated glomerular filtration rate) were performed before starting the VLCKD and after the 6 weeks of diet therapy with meal replacements. Body weight (BW), height, systolic and diastolic blood pressure, waist circumference (WC), thigh circumference (TC), and hip circumference (HC) were measured at the first visit (T0) and every two weeks thereafter (T2 and T4) until the end of the nutritional protocol (T6). To evaluate muscle mass and strength, a handgrip strength (HG) measurement and a chair stand test (CST) were performed in accordance with the ESWGOP2 report [4]. The HG was measured using a digital dynamometer (Dynx, Akern, Pontassieve, FI, Italy) at T0 and T6 [24]. All measurements were carried out with dominant and non-dominant arms and the highest value was recorded. The CST measured the amount of time needed for a patient to rise five times from a seated position without using arms: sarcopenia is determined if the patient takes more than 15 s, in accordance with the EWGSOP2 report [4]. To assess sarcopenic condition severity, the short physical performance battery (SPPB) was evaluated at the beginning of the protocol, which led to the exclusion of a severe sarcopenic condition (data not shown). Body composition, total body fat mass (FM), and fat-free mass (FFM) were also measured, using multifrequency bioelectrical impedance analysis (BIA, Human Im Touch, DS Medica S.r.l., Milan, Italy) at baseline and at the end of the protocol [25]. The Human Im Touch device records impedance at five frequencies (5, 10, 50, 100, and 250 kHz). During the BIA, patients were lying in a supine position. All measurements were performed on the patient’s right side. The four-surface standard tetra polar electrode technique on the foot and hand was used. Seven days after the beginning of the nutritional protocol, the patient started Interval Training (IT) twice a week. Due to the pandemic, physical exercise sessions were carried out via the Zoom platform with a personal trainer, and each session lasted 30–35 min. The required home-based equipment consisted of a stable chair with a backrest and without armrests, two bottles of water, and a towel. Each session of physical exercise was structured as follows: an initial warm up, which involved breathing exercises and stretching of the posterior chain, a second part based on functional exercises repeated for 30 s with a 15 s pause, a part focusing on proprioception and balance, and finally, a part focused on breathing.

## 3. Results

Variations in BW, systolic and diastolic blood pressure, WC, and HC are reported in Table 1: the combination of the VLCKD with IT led to a reduction in BW (110 vs. 94 kg), BMI (36.7 vs. 31.4 kg/m^2^), BP (135/85 vs. 120/80 mmHg), and WC (110 vs. 98 cm). As shown in Table 2, metabolic parameters were assessed by performing blood tests after the patient’s discharge from subintensive care (T-1), before starting VLCKD + IT (T0), and after 6 weeks (T6). Notably, a reduction in fasting insulin (12 vs. 9 μIU/mL), fasting glycemia (123 vs. 92 mg/dL), the HOMA index (3.6 vs. 2.0), creatinine (0.9 vs. 0.73 mg/dL), triglycerides (145 vs. 82 mg/dL), and LDL cholesterol (104 vs. 73 mg/dL) was observed from T0 to T6, with a parallel increase in HDL cholesterol (40 vs. 50 mg/dL) and vitamin D (23 vs. 29.6 ng/mL). Changes in electrolytes were not observed (data not shown). The average weight loss was nearly 14.5% of the patient’s initial weight, with a reduction in fat mass (53.2 vs. 38 kg) and a concomitant preservation of fat-free mass (56.8 vs. 56 kg), as reported in Figure 1 and Table 3. Improvement in muscle strength and physical performance, as measured by the HG and the CST, was observed at T6 (Figure 2 and Table 3).

## 4. Discussion

In this brief case report, we analyzed the effects of a 6-week VLCKD, combined with a physical exercise program, in a middle-aged woman affected by SO and dyslipidemia. As observed in a previous study, the VLCKD was effective as a dietary approach, determining a reduction in BW and fat mass, and skeletal muscle maintenance [26]. In general, skeletal muscle loss may occur, along with BW reduction, during a hypocaloric diet [27], and an increase in muscle proteolysis has been suggested to play a role in muscle mass reduction under calorie restriction [27]. Amino acid availability affects muscle proteolysis and protein synthesis, and diets designed for weight loss which have a high protein content have been shown to preserve lean mass [28,29]. 

Interestingly, supplementation with whey protein and leucine has been shown to maintain skeletal muscle mass in subjects with obesity following a hypocaloric diet and a resistance training program [30]. In general, VLCKD provides a total daily protein intake of 1.2 g (female) or 1.5 g (male) per kg of BW, which is expected to preserve muscle during weight loss. Accordingly, the present case report indicates that a VLCKD was able to reduce fat mass and maintain lean mass after 6 weeks of dietary therapy, which remarkably confirms, and extends to a longer observation period, the protective effects against sarcopenia, as described by Merra et al. after 3 weeks of dietary intervention [26]. A recent study has shown that aerobic and resistance exercises are able to counteract a decrease in lean mass in obese subjects who are dieting, indicating beneficial effects on physical function, as shown by the physical performance test, Functional Status Questionnaire, one-repetition maximum strength test, and peak oxygen consumption measurements [31]. Resistance training, and a diet with an adequate protein content, have also been suggested as valuable tools for preserving muscle mass [27]. As previously observed by Galbreath et al., a dietary protein intake of 1.2 g/kg/d, which is usually recommended for a ketogenic diet, combined with a resistance-based exercise, was able to reduce body weight and fat mass in overweight women, without a significant decrease in fat-free mass [32].

Accordingly, in the present case report, BIA measurements showed that fat-free mass was preserved after 6 weeks of VLCKD combined with physical exercise (Figure 1). In addition to muscle mass preservation, we observed that both the HG and CST tests (Figure 2) showed an increase in physical performance, despite a reduced fat mass (Figure 1). In addition to the role of ketone bodies as energy substrates, these molecules also modulate several cellular processes, such as inflammation, oxidative stress, and gene expression regulation [33]; importantly, β-hydroxybutyrate (BHB) has been shown to exert anticatabolic effects on human skeletal muscle [34]. As mentioned above, resistance training could also contribute to maintain muscle mass, improve physical strength [35], and, in parallel, counteract the excessive expansion of adipose tissue [36] (Figure 1). Physical exercise has been shown to exert direct effects on adiponectin expression by increasing its circulating levels, with potential protective effects on muscle mass and function [37]. A limitation of the study is represented by the short intervention period (6 weeks), which raises the need to investigate the long-term effects of ketogenic diets in subjects with SO. he subject described in this case report followed a dietary intervention combined with exercise after hospital discharge for bilateral interstitial pneumonia due to COVID-19. Obesity and related comorbidities, including glycemic disorders and sarcopenia, may influence vulnerability to post-COVID-19 physical deterioration [38]. In a retrospective study, a significant association between reduced muscle mass and the onset of complications from COVID-19 was observed, confirming that muscle mass depletion was predictive of an unfavorable clinical outcome in COVID-19 patients during the first pandemic wave [39]. Indeed, SARS-CoV-2 infection determines physical decline, with reduced appetite, chronic cardiorespiratory symptoms, social isolation, and reduced physical activity [40], leading to sustained post-infection sequelae, and the severe worsening of health status and quality of life, which is a condition known as “long COVID” [41]. 

Our results suggest that a VLCKD, together with physical training, is effective in protecting muscle mass and physical performance, and represents an attractive physical and metabolic rehabilitation approach against long COVID. 

## 5. Conclusions

The long-term safety of very-low-carbohydrate dietary regimens still raises concerns, despite several recent studies supporting the beneficial impact of a VLCKD on adipose and skeletal muscle metabolism in the management of obesity [23]. In patients with obesity, this dietary approach also improves lipid and glycemic profiles, with proven cardiometabolic protective effects. This case study suggests that a VLCKD, combined with physical training, reduces adipose depots and preserves fat-free mass, improving muscle strength during weight loss. In addition, this combined approach, through muscle mass preservation, might protect against post-COVID-19 metabolic derangements [42]. However, investigation of the metabolic effects of a VLCKD, in combination with physical exercise, requires additional studies. 

## Figures and Tables

**Figure 1 healthcare-10-00573-f001:**
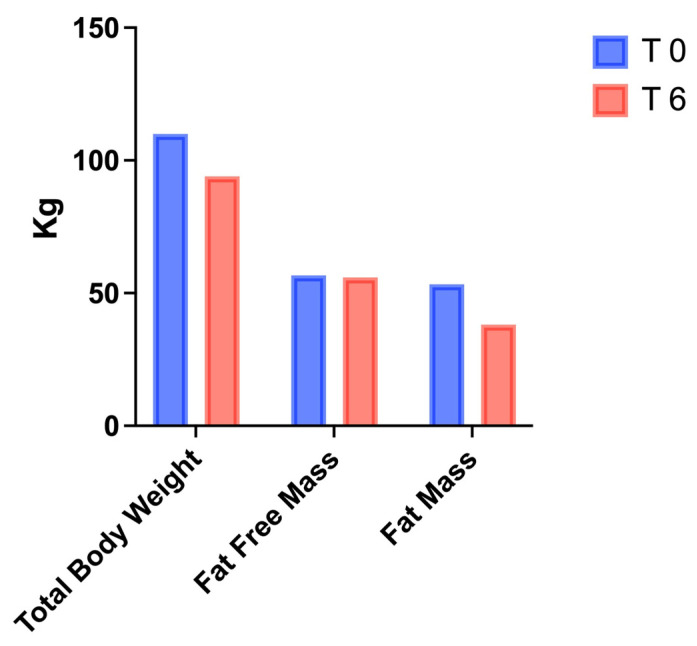
Body composition analysis was performed using BIA at the beginning (T0) and at the end (T6) of VLCKD+IT.

**Figure 2 healthcare-10-00573-f002:**
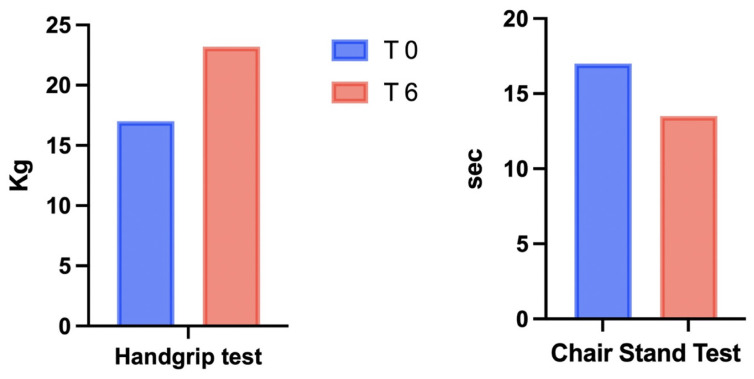
Physical performance measured by the handgrip strength test and chair stand test at the beginning (T0) and at the end (T6) of VLCKD+IT.

**Table 1 healthcare-10-00573-t001:** Anthropometric measurements of the patient at baseline (T0), every 2 weeks (T2 and T4), and at the end (T6) of VLCKD+IT.

	T 0	T 2	T 4	T 6
Height (cm)	1.73			
Weight (kg)	110	105	99	94
Body Mass Index (kg/m^2^)	36.7	35	33	31.4
Waist Circumference (cm)	110	106	102	98
Hips Circumference (cm)	137	134	138	122
Systolic Body Pressure (mmHg)	135	130	120	120
Diastolic Body Pressure (mmHg)	85	80	80	80

**Table 2 healthcare-10-00573-t002:** Blood test results after discharge (T-1), at baseline (T0), and after 6 weeks (T6) of VLCKD+IT.

	T -1	T 0	T 6
Fasting Glycemia (mg/dL)	108	123	92
Fasting Insulin (μUI/mL)	12	12	9
HOMA Index	3.6	3.6	2.0
Creatinine (mg/dL)	0.82	0.9	0.73
eGFR (ml/min)	81	65	93
AST (U/L)	30	65	26
ALT (U/L)	80	39	24
Total Cholesterol (mg/dL)	137	185	140
LDL Cholesterol (mg/dL)	81	104	73
HDL Cholesterol (mg/dL)	33	40	50
Triglycerides (mg/dL)	112	145	82
Vitamin D (ng/mL)	20	23	29.6
TSH (μUI/mL)	0.5	2.1	2.7

**Table 3 healthcare-10-00573-t003:** Characteristics of body composition and physical performance parameters at baseline (T0) and after 6 weeks (T6) of VLCKD+IT.

	T 0	T 6
Fat-Free Mass (kg)	56.8	56
Fat Mass (kg)	53.2	38
Fat Free Mass (%)	51.6	59.6
Fat Mass (%)	48.4	40.4
Chair Stand Test (sec)	17	13.5
Handgrip Test (kg)	17	23.2

## Data Availability

Not applicable.

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
