# Peer review of "VLCKD in Combination with Physical Exercise Preserves Skeletal Muscle Mass in Sarcopenic Obesity after Severe COVID-19 Disease: A Case Report"

_healthcare, 2022, doi:10.3390/healthcare10030573_

Round 1
Reviewer 1 Report
The paper is About a case study in very low carbo ketogenic diet in an obese middle age female subject with associate co-morbodities.
The paper Show the efficacy of a ketogenic diet in ameliorating the Health and the objective ematological profile of the subject .
English language is now fine.
The paper has been improved since the first review. It can be an interesting paper for the clinical practice.
Weaknesses:
There is no comparison from the literarure, neither in the introduction or in the discussion sections, with similar cases employing different Kinds of dietary interventions than VLCKD.
In summary the paper can be interesting for the clinical practice within the limits of the case reports studies.
Line 330 (table 3 title) has a typo error (miss: „Table 3“) or I can´t see it in my pdf.
the hand dynamometer is Dynx not Dynex. Please report the reference for the validation of the device. Same for BIA device
Author Response
Manuscript ID: healthcare-1600951
Title: “VLCKD In Combination With Physical Exercise Preserves Skeletal Muscle
Mass In Sarcopenic Obesity After Severe COVID-19 disease: A Case Report”
Dear Editor,
We thank the expert referees for the constructive comments on our manuscript.
We modified the manuscript along the requested issues.
All the corrections and minor changes throughout the manuscript have been highlighted in yellow.
Reviewer 1
We thank the reviewer for the comments.
Point 1: We have now added a title for Table 3: “Characteristics of body composition and physical performance parameters at baseline (T0) and after 6 weeks (T6) of VLCKD+IT.”
Point 2: As suggested, we have now added the references for the validation of the dynamometer (reference n 24) and BIA (reference n 25) device.
We hope that the manuscript in its present form is acceptable for publication.
Kind regards,
Prof. Massimiliano Caprio

Reviewer 2 Report
The authors have addressed my previous comments. However, a few clarifications are suggested:
- Page 3 lines 246-247 can add and clarify "...were measured at the first visit (T0) and every two weeks (T2 and T4) until the end of nutritional protocol up to 6 weeks (T6)."
- Table 3 is missing the title in the Figure Legend.
Author Response
Manuscript ID: healthcare-1600951
Title: “VLCKD In Combination With Physical Exercise Preserves Skeletal Muscle
Mass In Sarcopenic Obesity After Severe COVID-19 disease: A Case Report”
Dear Editor,
We thank the expert referees for the constructive comments on our manuscript.
We modified the manuscript along the requested issues.
All the corrections and minor changes throughout the manuscript have been highlighted in yellow.
Reviewer 2
We thank the reviewer for the precious comments.
Point 1: As suggested, we have now added “Body weight (BW), height, systolic and diastolic blood pressure, waist circumference (WC), thigh circumference (TC) and hip circumference (HC) were measured at the first visit (T0) and every two weeks (T2 and T4) until the end of nutritional protocol (T6)”
Point 2: Now we added the title for Table 3 “Table 3. Characteristics of body composition and physical performance parameters at baseline (T0) and after 6 weeks (T6) of VLCKD+IT.”
We hope that the manuscript in its present form is acceptable for publication.
Kind regards,
Prof. Massimiliano Caprio

This manuscript is a resubmission of an earlier submission. The following is a list of the peer review reports and author responses from that submission.
Round 1
Reviewer 1 Report
The paper is a case study of an obese woman who undergone a diet/physical activity program after being hospitalized for covid 19.
The topic is interesting being obesity a co-morbidity worsening several clinical situations, including covid. Also, the association of a ketogenic diet and exercise in a therapeutic approach that can be interesting for clinical practice.
However, the results are quite obvious: diet and exercise reduced body weight and increased fitness in the subjects.
Where is the novelty or the originality of the study is not clear. Overall, the paper does not add nothing to the existing knowledge on the topic and is more a clinical report than a scientif paper.
Reviewer 2 Report
The manuscript is well-written and described. However, changes are suggested:
- Two acronyms are not described, DEXA or BIA.
- The information described in Page 3 lines 117-122 can be presented in a Table for better understanding and visualization.
- Is the information described in Page 3 lines 117-126, describing T6 vs T0 or T0 vs T6? The sentence says: "Following nutritional therapy and physical activity, a reduction of BW (94 vs 110 Kg)...". In this form the results are shown an increase.This should be clarified.
- The sentence in Page 3 lines 124-126 says an improvement was observed in Figure 2, this is only true for Handgrip Test, not Chair Stand Test.
- How does the manuscript justify that the diet improves after COVID-19 exposure if the results are not showing a previous measurement before COVID-19?
- Could the medications taken by the patient have an effect on the results?
- The Case Description mentions that this patient was closely and periodically monitored through physical examination. However, this results are not presented periodically in the manuscript.
Reviewer 3 Report
The case reported in the manuscript presented dietary interventions and exercise for muscle mass preservation in a patient after severe COVID 19 disease. The authors choose a very low calorie ketogenic diet, but in the introduction and discussion they point to studies demonstrating the effectiveness of the ketogenic diet for decreasing inflammation, weight loss, and other. Thus, it was not clear why they choose a very low-calorie combined with ketogenic diet. This explanation should be in the manuscript